# Preoperative Prediction of New Vertebral Fractures after Vertebral Augmentation with a Radiomics Nomogram

**DOI:** 10.3390/diagnostics13223459

**Published:** 2023-11-16

**Authors:** Yang Jiang, Wei Zhang, Shihao Huang, Qing Huang, Haoyi Ye, Yurong Zeng, Xin Hua, Jinhui Cai, Zhifeng Liu, Qingyu Liu

**Affiliations:** 1Department of Radiology, The Seventh Affiliated Hospital, Sun Yat-sen University, Shenzhen 518000, China; jiangy279@mail2.sysu.edu.cn (Y.J.); zhangwei3@sysush.com (W.Z.); caijinhui@sysush.com (J.C.); 2Department of Radiology, The Fifth Affiliated Hospital, Sun Yat-sen University, Zhuhai 519000, China; m15285361659@163.com; 3Department of Endocrinology, The Seventh Affiliated Hospital, Sun Yat-sen University, Shenzhen 518000, China; huangq77@mail2.sysu.edu.cn; 4Department of Radiology, The Fourth Affiliated Hospital, Guangzhou Medical University, Guangzhou 511300, China; 2021689031@gzhmu.edu.cn; 5Department of Radiology, Huizhou Central People’s Hospital, Huizhou 516000, China; zeng_yurong@163.com; 6Department of Neurology, The First Affiliated Hospital, Wenzhou Medical University, Wenzhou 325000, China; huaxin04157915@163.com

**Keywords:** osteoporosis, vertebral fracture, vertebral augmentation, prediction, radiomics

## Abstract

The occurrence of new vertebral fractures (NVFs) after vertebral augmentation (VA) procedures is common in patients with osteoporotic vertebral compression fractures (OVCFs), leading to painful experiences and financial burdens. We aim to develop a radiomics nomogram for the preoperative prediction of NVFs after VA. Data from center 1 (training set: *n* = 153; internal validation set: *n* = 66) and center 2 (external validation set: *n* = 44) were retrospectively collected. Radiomics features were extracted from MRI images and radiomics scores (radscores) were constructed for each level-specific vertebra based on least absolute shrinkage and selection operator (LASSO). The radiomics nomogram, integrating radiomics signature with presence of intravertebral cleft and number of previous vertebral fractures, was developed by multivariable logistic regression analysis. The predictive performance of the vertebrae was level-specific based on radscores and was generally superior to clinical variables. RadscoreL2 had the optimal discrimination (AUC ≥ 0.751). The nomogram provided good predictive performance (AUC ≥ 0.834), favorable calibration, and large clinical net benefits in each set. It was used successfully to categorize patients into high- or low-risk subgroups. As a noninvasive preoperative prediction tool, the MRI-based radiomics nomogram holds great promise for individualized prediction of NVFs following VA.

## 1. Introduction

Osteoporosis is a prevalent degenerative skeletal disease marked by low bone mass and microarchitectural deterioration of bone tissue, causing skeletal fragility and high fracture risk [1]. By 2050, the proportion of osteoporotic fractures in all fractures will increase to 50%, and osteoporotic fractures are also one of the leading causes of disability and death in the elderly [2,3]. Importantly, osteoporotic vertebral compression fractures (OVCFs) are the predominant fractures linked with skeletal fragility, and they result in significant morbidity, mortality, and other adverse health consequences [4,5].

Considering the significant morbidity that is linked to OVCFs, it is not surprising that there is a great deal of enthusiasm surrounding vertebral augmentation (VA). Vertebral augmentation, including vertebroplasty (VP) or balloon kyphoplasty (BKP), is a minimally invasive procedure designed to “fix” acute vertebral fractures (VFs), thereby facilitating biomechanical stability and functional recovery [6]. However, several studies have reported an increased risk of new vertebral fractures (NVFs) associated with VA or additional VFs occurring sooner in patients who undergo VA procedures than in patients with nonsurgical management [7,8]. Approximately 20% of patients with OVCFs are estimated to experience NVFs following VA, with most occurring within one year after the procedure, resulting in substantial patient suffering [9,10,11]. Therefore, identifying the population that are at a high risk of NVFs within one year prior to VA can aid clinicians with individualized treatment and management of patients with VFs [12].

The determination of VFs can be assisted by imaging modalities such as magnetic resonance imaging (MRI) and computed tomography (CT). In routine clinical practice, MRI has proven to be the most appropriate diagnostic modality to identify acute benign vertebral compression fractures caused by osteoporosis due to its excellent soft tissue contrast [13,14]. Generally, MRI examinations are performed ahead of the VA procedure, and it may be possible for clinicians to treat and monitor patients at risk for NVFs more effectively if they can make use of preoperative MRI data [15].

In recent years, research involving radiomics analysis has rapidly grown. Radiomics refers to the high-throughput extraction of numerous quantitative features from medical images, which provides objective information that would be challenging for the human eye to quantify. Radiomics analysis has been used in several previous studies to assess osteoporosis and VFs with promising results [16,17]. We hypothesized that radiomics analysis of MRI images would provide more quantifiable and objective information on the spatial heterogeneity within fractured vertebrae than visual evaluation, potentially improving the accuracy of the model for predicting VFs.

The incidence of fractures after VA procedures varies among vertebrae, and intrinsic characteristics within vertebrae, such as volumetric bone mineral density (BMD), have been revealed to display variation at different levels of the spine [18,19]. All of these findings imply that there exists heterogeneity among vertebrae, highlighting that inappropriate vertebral region of interest selection may potentially limit the performance of prediction models [20]. MRI-based radiomic features have been reported to reflect intrinsic characteristics and may represent a possible avenue for exploring the heterogeneity among vertebrae [21].

Therefore, in this study, we aimed to compare the level-specific predictive performance of vertebrae for NVFs using radiomics scores and then develop an MRI-based radiomics nomogram as a clinical application tool for preoperative prediction of NVFs that occur within one year following VA.

## 2. Materials and Methods

### 2.1. Study Participants

This retrospective multi-institutional study was reviewed and approved by the institutional review boards of all participating institutions, and the requirement to obtain written informed consent was waived.

The T1-weighted MRI images and clinical data of patients who underwent the VA procedure at center 1 between September 2013 and May 2020 and center 2 between June 2013 and September 2020 were reviewed (Figure 1). The inclusion criteria were as follows: (1) female patients aged >55 years and male patients aged >60 years; (2) patients diagnosed with acute OVCFs who then received VA procedures; and (3) patients with complete clinical and imaging data. The exclusion criteria were as follows: (1) patients with fractures caused by high-energy trauma, infection, or tumor; (2) patients who died during the follow-up period or declined to follow-ups; and (3) patients with secondary osteoporosis due to hyperparathyroidism, inflammatory bowel disease, or renal failure.

Information from VA procedures (surgical procedures, location of treated vertebrae, and number of treated vertebrae) was collected from the medical records, as well as baseline clinical data (age, sex, body mass index (BMI), and smoking status). MRI findings, such as the presence of an intravertebral cleft (IVC) and previous VF, and BMD were available for patients who performed preoperative MRI and dual-energy X-ray absorptiometry (DXA).

All enrolled patients were followed up every three months after the surgery until the end of the one-year follow-up period or the occurrence of NVFs, whichever came first. If patients presented symptoms suggestive of NVFs during the follow-up period, such as acute back pain, they underwent an MRI examination immediately. During the last follow-up visit, postoperative spinal MRI was performed to determine whether NVFs had occurred. Patients who were diagnosed with acute NVFs had bone marrow edema present on preoperative spinal MRI. In total, 219 patients from center 1, including 88 patients with NVFs and 131 patients without NVFs, were assigned to the training and internal validation sets in a 7:3 ratio. Forty-four patients from center 2 were included in the external validation set, including fifteen patients with NVFs and twenty-nine patients without NVFs.

### 2.2. Image Acquisition and Radiomics Feature Extraction

Prior to the surgery, all patients underwent an MRI examination, and the MRI image acquisition settings for each center are presented in Appendix A.

The radiomics workflow is shown in Appendix A. The volume of interest (VOI) of T11-L5 vertebrae was manually delineated from preoperative spinal MRI using the publicly available 3D Slicer software version 4.13.0 and fractured vertebrae were excluded.

Including repeatable features in prediction models can be crucial for ensuring model generalizability, and we followed a three-step image preprocessing procedure to identify repeatable radiomic features. First, image inhomogeneity was corrected using the N4 bias field correction algorithm. Next, to standardize the voxel spacing, all MRI images were resampled to a voxel size of 1 × 1 × 1 mm^3^; we then used Z Score to improve the reproducibility of multicenter radiomic studies on MRI datasets. A total of 1130 radiomics features were extracted from each VOI using the PyRadiomics platform (version 3.0.1) implanted in Python software (version 3.7.1).

### 2.3. Radiomics Feature Selection and Radiomics Score Construction

To evaluate whether there were significant differences in radiomics features between patients with NVFs and those without NVFs, Mann–Whitney U tests were applied. Additionally, the least absolute shrinkage and selection operator (LASSO) algorithm, a powerful machine learning method for regression with high-dimensional data, was used to further screen the radiomics features with a *p* value less than 0.05 from the Mann–Whitney U tests. A fivefold cross-validation was conducted on the training set to tune the penalty parameter. The level-specific vertebral radiomics score (radscore) was calculated by linearly combining the final included radiomics features and weighting them according to their respective LASSO coefficients.

### 2.4. Construction and Evaluation of a Predictive Radiomics Nomogram

In the three datasets, the discrimination of the radscores was evaluated using the area under the curve (AUC) of the receiver operator characteristic. Moreover, discrimination of other predictive variables of NVFs after VA was also assessed and validated and compared with the radscores of different levels of the vertebrae. The radiomics signature was constructed by the radscores of the vertebrae with the best predictive performance.

Each candidate predictive variable, including the radiomics signature and clinical variables, was assessed using a univariate logistic regression algorithm in the training set. A subsequent multivariate logistic regression analysis was conducted with variables with a *p* value less than 0.05 in the univariate analysis. Based on the multivariate logistic regression analysis incorporating the selected factors, a predictive radiomics nomogram was constructed.

The AUC was used to determine the nomogram discrimination, and the validity of the nomogram was evaluated using the calibration curve. To determine the clinical usefulness of the nomogram, decision curve analysis (DCA) was performed by calculating the net benefits at different threshold probabilities. Based on the cutoff score of the nomogram, the set of all patients was then divided into a low-risk group and a high-risk group.

### 2.5. Statistical Analysis

All statistical analyses were performed using free R (version 4.1.3) and SPSS (version 26.0). The normally distributed continuous variables were compared using the t test, while nonnormally distributed continuous variables were compared using the Mann–Whitney U test. The 2-group categorical variables were compared using the χ^2^ test. The cumulative incidence curves were estimated by using the Kaplan–Meier method and compared with the log-rank test. The ROC curves were plotted using the “pROC” package. Logistic regression, nomogram construction, and calibration plots were performed with the “rms” package. DCA was performed with the function “ggDCA”. For all analyses, *p* < 0.05 was considered statistically significant, and all tests were 2-tailed.

## 3. Results

### 3.1. Characteristics of the Study Sets

Table 1 shows detailed baseline characteristics of the training set (153 participants; median [interquartile range (IQR)] age, 76 [68, 82] years), internal validation set (66 participants; median (IQR) age, 73 [67, 81] years), and external validation set (44 participants; median (IQR) age, 73 [70, 80] years). Significant statistical differences in IVC were observed across all study sets of patients with and without NVFs.

Overall, we analyzed MRI images of 1436 vertebrae in the T11–L5 segments, and 229, 178, 153, 208, 208, 220, and 240 vertebrae were included in the T11, T12, L1, L2, L3, L4, and L5 segments, respectively. The median time to fracture occurrence was 6 months for the training set, 5.5 months for the internal validation set, and 6 months for the external validation set.

### 3.2. Construction and Validation of the Radiomics Score

In total, 1130 radiomics features were extracted from each VOI of the vertebrae on T1-weighted MR images. Then, Mann–Whitney U tests and the LASSO algorithm confirmed the final key features of each vertebra. The formula and distribution for the radscore of each vertebra in all datasets are presented in Appendix A.

The radscores of patients with NVFs were usually higher than that of patients without NVFs. Among all these radscores, RadscoreL2 indicated the most favorable prediction of NVFs with an AUC of 0.850 (95% confidence interval (CI), 0.773–0.909) in the training set, which was validated in the internal validation set with an AUC of 0.783 (95% CI, 0.644–0.887) and in the external validation set with an AUC of 0.751 (95% CI, 0.587–0.876) (Appendix A). The radiomics signature, constructed by the radscore of L2 and adjacent vertebrae, demonstrated good predictive performance similar to RadscoreL2 with an AUC of 0.853 (95% CI, 0.787–0.905) in the training set, 0.809 (95% CI, 0.693–0.895) in the internal validation set, and 0.798 (95% CI, 0.649–0.904) in the external validation set.

We also compared the predictive performance of the radscore with that of clinical risk variables for NVFs after VA. In the three datasets, age, IVC, and number of previous VFs had moderate predictive performance (AUC > 0.6) among all clinical risk variables, while the predictive performance of other variables was poor (AUC > 0.5) (Appendix A). Most of the radscores for constructed vertebrae showed better predictive performance than clinical risk variables.

### 3.3. Prediction Radiomics Nomogram Development and Validation

In the training set, four variables, including age, radiomics signature, IVC, and number of previous VFs, were found to be significant at a level of *p* < 0.05 based on the univariate logistic regression algorithm (Table 2). Among them, three variables, including the number of previous VFs, IVC, and radiomics signature, were selected using the multivariate logistic regression algorithm. Then, a radiomics nomogram incorporating these three variables was developed based on the multivariate logistic regression model (Figure 2).

The radiomics nomogram showed favorable discrimination with an AUC of 0.886 (95% CI, 0.834–0.938) in the training set (Figure 3A). The calibration curve demonstrated excellent calibration of the radiomics nomogram (Figure 4A). Furthermore, the Hosmer–Lemeshow test yielded a nonsignificant statistic (*p* = 0.627), indicating a robust fit. As depicted in Figure 3B, the significant discrimination of the radiomics nomogram was confirmed in the internal validation set (AUC (95% CI), 0.834 (0.729–0.940)). In addition, the performance was confirmed in an external validation set with an AUC of 0.867 (95% CI, 0.752–0.982). As shown in Figure 4B,C, there was good consistency between the actual and nomogram-predicted NVF rates in the two validation sets, with nonsignificant *p* values (0.639 and 0.576, respectively) derived from the Hosmer–Lemeshow test.

DCA demonstrated in the training set that predicting NVFs within one year after the VA procedure using the radiomics nomogram adds a greater net benefit than treating all patients or treating none for a wide range of threshold probabilities, indicating the radiomics nomogram as a clinically useful tool (Figure 5A). Additionally, similar findings were also observed when DCA was applied to both the internal and external validation sets (Figure 5B,C).

### 3.4. Risk Stratification

With total points of 55.49 as the cutoff score of the training set, the preoperative radiomics nomogram identified low-risk and high-risk categories of patients after VA. In the training set, the cumulative 1-year occurrence incidences of NVFs were 68.35% and 13.51%, respectively, for high-risk patients and low-risk patients (*p* < 0.001, log-rank test, Figure 6A). In the two validation sets, two distinct prognostic strata were also confirmed (*p* < 0.001, log-rank test, Figure 6B,C).

## 4. Discussion

Preventing and treating NVFs following VA pose a considerable challenge for clinicians because NVFs commonly occur within one year after surgery and can lead to pain and financial burden. In the present study, we developed an MRI-based radiomics nomogram combining the radiomics signature, IVC, and number of previous VFs for individualized preoperative prediction of NVFs after VA that achieved high discrimination performance, favorable calibration, and good clinical usefulness. In addition, the radiomics nomogram could be used to successfully categorize patients who underwent VA procedures into two risk subgroups. The similar performance of the radiomics nomogram in the internal validation set and the external validation set that were obtained from other institutions suggested the reproducibility and reliability of the proposed nomogram.

As osteoporosis progresses, bone density decreases and bone microstructure deteriorates, resulting in an increased risk of fractures. Importantly, the primary goal of osteoporosis treatment is to reduce the risk of clinical fractures, particularly OVCFs. DXA is commonly used in clinical practice to measure bone loss and determine the risk of osteoporosis fracture. However, DXA is widely underused in some countries, and BMD, which is derived from DXA, can only reveal bone mass and cannot reflect bone quality [22,23]. It has been shown that numerous patients with fragility fractures were not diagnosed with osteoporosis based on BMD [24]. In addition, the predictive performance of BMD in our study (AUC ≤ 0.575 in three sets) is limited, and other tools for osteoporotic fracture prediction warrant further investigation.

IVC, one of the included variables in the radiomics nomogram, refers to a radiolucent zone that is filled with liquid or gas within a fractured vertebra. The presence of an IVC may indicate delayed tissue mineralization caused by decreased osteoblast activity and excessive tissue absorption due to increased osteoclast activity [25]. IVC has been reported as an independent risk factor and potential predictor for NVFs following VA [26,27]. A history of previous VFs is also an important risk factor for developing NVFs. The risk of developing NVFs is approximately five times higher in patients who have previously experienced a fracture, and more than 20% of patients will experience an additional VF within one year of their first fracture [28].

Radiomics features can capture intrinsic characteristics, such as lesion heterogeneity, by the conversion of medical images into high-dimensional data that can be mined for analysis [29]. As a result, radiomics-based tools have been developed to enhance diagnostic accuracy for osteoporosis and improve fracture prediction [30,31]. In the current study, the discrimination of radiomics scores was favorable (AUC ≥ 0.792) and consistently superior to clinical variables (AUC ≤ 0.677) in the training set and also in the validation sets. These results suggest that the radiomics score has the potential to be an effective tool for identifying patients at high risk of VFs following VA.

Notably, level-specific vertebrae not only differ in the related incidence of VFs but also present disparities in intrinsic characteristics, such as BMD and fat fraction [18,32,33]. These intrinsic characteristics have been established to be associated with osteoporotic fractures and can assist with the identification of such fractures [34,35]. The observed inconsistency reflects the heterogeneity among vertebrae and thereby emphasizes the need for extensive investigation into this phenomenon. Radiomics features were extracted, and radiomics scores were constructed for each level of vertebrae separately in this study. In accordance with previous reports, wavelet-based features achieved the highest weights in the current study among all selected radiomics features [36,37]. These features, which may further reflect the spatial heterogeneity of a lesion at multiple scales, can provide more detailed information that is complementary to visual assessment. This finding may provide insight into the level-specific predictive performance of vertebrae based on radiomics scores. Furthermore, the predictive performance was highest at the thoracolumbar junction, especially at the L2 vertebrae, which may be attributed to the higher susceptibility to and earlier initiation of bone loss in this region, and these subtle internal changes within the vertebrae are well captured by the radiomic features [38].

Our radiomics nomogram surpasses the limitation of previous studies that failed to differentiate between various levels of vertebrae, resulting in superior predictive performance (AUC (95% CI), 0.886 (0.834–0.938) vs. 0.810 (0.773–0.843), respectively) [20]. Moreover, the three selected variables included in the nomogram can be easily obtained in clinical practice without any additional burden.

Treatment strategies vary depending on the fracture risk group. In general, antiresorptive agents are recommended for treating osteoporosis and reducing the risk of fractures. Indeed, the best course of action for patients who are at high risk of fracture is to initiate treatment with an anabolic agent as soon as possible [39,40]. Our radiomics nomogram effectively identifies high-risk individuals among all patients, thereby contributing to better treatment outcomes for this population.

Additionally, the present study has some noteworthy points and limitations. First, while the number of patients in the external validation set satisfies the fundamental analytical requisites of radiomics modeling, more external data will be required in the future to further validate our radiomics nomogram [41]. Second, manual vertebral segmentation is time-consuming and laborious, which may lead to potential limitations on the application of the nomogram. Automated vertebral segmentation based on deep learning is convenient and has satisfactory accuracy, possibly providing a solution for this issue [42]. Third, previous studies have indicated that intervertebral discs or paravertebral muscles may also exert an influence on the occurrence of VFs [43,44]. Therefore, incorporating more information into prediction models could enable accurate fracture prediction and represents a promising direction for future research.

## 5. Conclusions

The present study demonstrates the added value of applying radiomics features extracted from MRI images in the preoperative prediction of NVFs that occur within one year after the VA procedure, and vertebrae have level-specific predictive performance based on radiomics scores. The presented radiomics nomogram has the potential to serve as a noninvasive tool for individualized preoperative risk prediction of NVFs with favorable discrimination, calibration, and clinical usefulness, optimizing clinical treatment and routine management.

## Figures and Tables

**Figure 1 diagnostics-13-03459-f001:**
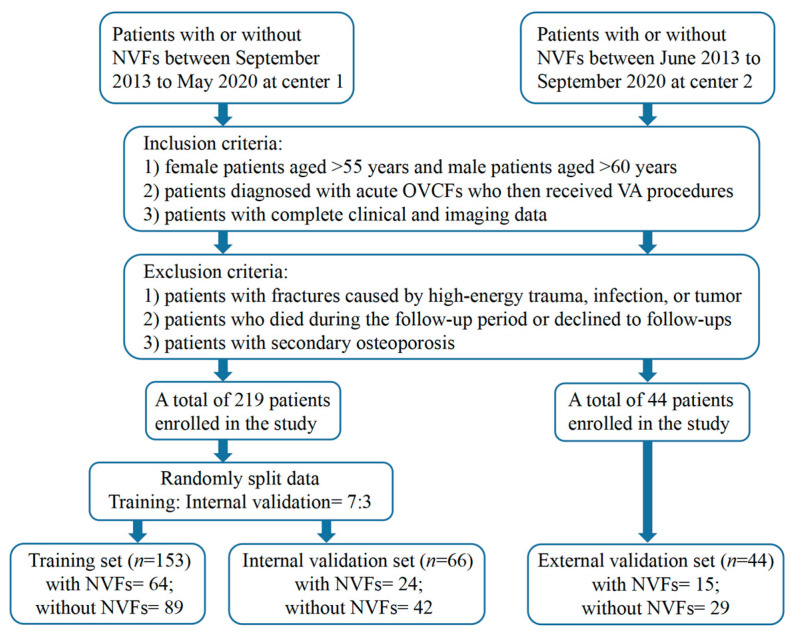
Flowchart of patient recruitment. Center 1, The Fourth Affiliated Hospital of Guangzhou Medical University; Center 2, Huizhou Central People’s Hospital.

**Figure 2 diagnostics-13-03459-f002:**
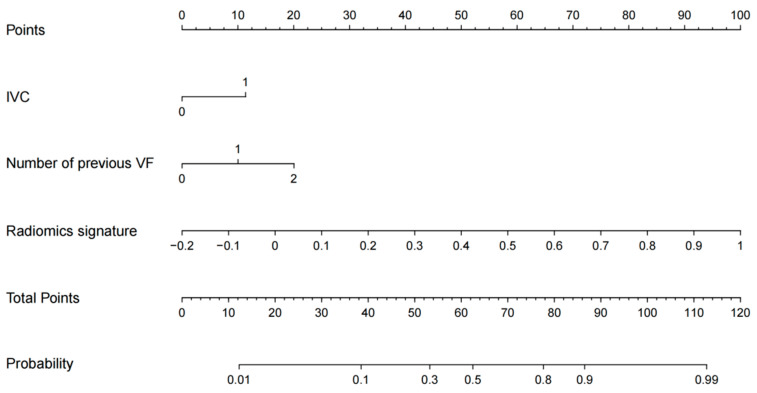
The MRI-based radiomics nomogram for NVFs prediction.

**Figure 3 diagnostics-13-03459-f003:**
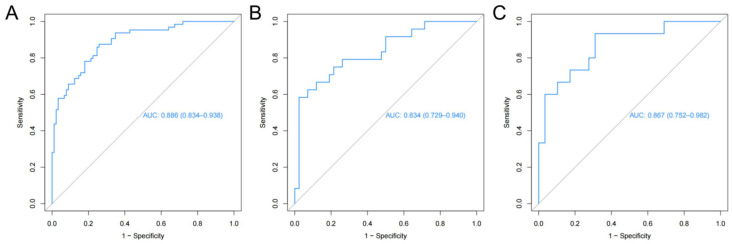
Receiver operating characteristic curves of the radiomics nomogram. (**A**) Training set, (**B**) internal validation set, (**C**) external validation set.

**Figure 4 diagnostics-13-03459-f004:**
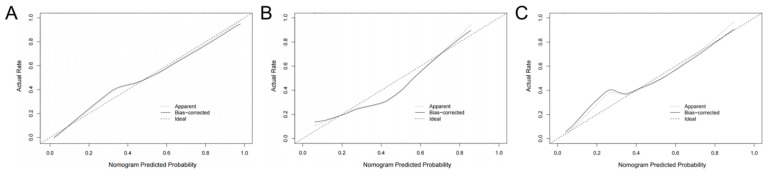
Calibration curves of the radiomics nomogram. (**A**) Training set, (**B**) internal validation set, (**C**) external validation set.

**Figure 5 diagnostics-13-03459-f005:**
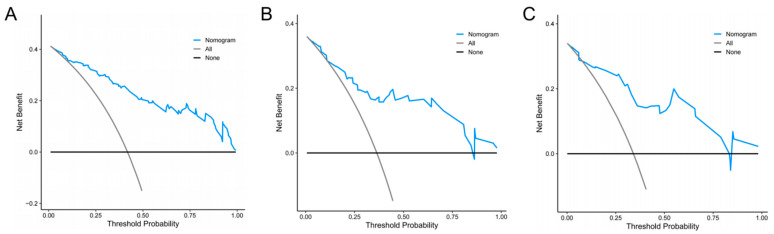
Decision curve analyses for the radiomics nomogram. The blue line represents the radiomics nomogram. The gray line represents the hypothesis that all patients had NVFs. The black line represents the hypothesis that no patients had NVFs. (**A**) Training set, (**B**) internal validation set, (**C**) external validation set.

**Figure 6 diagnostics-13-03459-f006:**
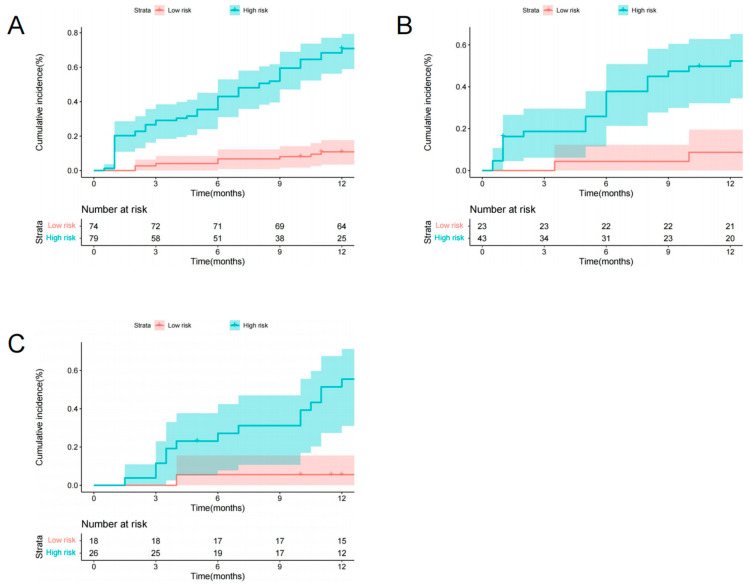
Graphs show cumulative incidence of NVFs according to two risk strata defined by the radiomics nomogram in the training (**A**), internal validation (**B**), and external validation (**C**) sets.

**Table 1 diagnostics-13-03459-t001:** Baseline patient characteristics (*n* = 263).

	Training Set (*n* = 153)			Internal Validation Set (*n* = 66)			External Validation Set (*n* = 44)		
Characteristic	without NVFs	with NVFs	*p*-Value	without NVFs	with NVFs	*p*-Value	without NVFs	with NVFs	*p*-Value
Age, yr, median	74 (66–82)	79 (72–82)	0.016	73 (65–81)	75 (69–81)	0.169	73 (70–76)	77 (71–82)	0.124
BMI (kg/m^2^)	23.09 ± 1.78	23.26 ± 1.68	0.555	23.35 ± 1.72	23.24 ± 1.90	0.817	23.15 ± 1.98	23.19 ± 2.08	0.958
BMD T-score	−3.16 ± 0.61	−3.24 ± 0.65	0.426	−3.13 ± 0.54	−3.20 ± 0.60	0.592	−3.20 ± 0.74	−3.44 ± 0.54	0.278
Sex									
Male	25 (28.1)	23 (35.9)	0.302	12 (28.6)	8 (33.3)	0.686	8 (27.6)	3 (20.0)	0.722
Female	64 (71.9)	41 (64.1)		30 (71.4)	16 (66.7)		21 (72.4)	12 (80.0)	
Smoking									
Absent	70 (78.7)	53 (82.8)	0.523	33 (78.6)	20 (83.3)	0.755	26 (89.7)	12 (80.0)	0.394
Present	19 (21.3)	11 (17.2)		9 (21.4)	4 (16.7)		3 (10.3)	3 (20.0)	
Surgical procedure									
VP	57 (64.0)	46 (71.9)	0.308	25 (59.5)	17 (70.8)	0.358	20 (69.0)	11 (73.3)	0.763
BKP	32 (36.0)	18 (28.1)		17 (40.5)	7 (29.2)		9 (31.0)	4 (26.7)	
IVC									
Absent	78 (87.6)	41 (64.1)	0.001	39 (92.9)	17 (70.8)	0.029	27 (93.1)	10 (66.7)	0.036
Present	11 (12.4)	23 (35.9)		3 (7.1)	7 (29.2)		2 (6.9)	5 (33.3)	
Number of treated vertebra									
1/2/3/4	70/16/3/0	52/5/4/3	0.046	35/6/1/0	16/4/2/2	0.183	26/3/0/0	12/1/2/0	0.235
Location of treated vertebra									
non-TL-Junction	25 (28.1)	27 (42.2)	0.069	17 (40.5)	12 (50)	0.453	9 (31.0)	5 (33.3)	0.877
TL-Junction	64 (71.9)	37 (57.8)		25 (59.5)	12 (50)		20 (69.0)	10 (66.7)	
Number of previous VF									
0/1/2	62/20/7	24/23/17	<0.001	24/11/7	8/8/8	0.141	19/9/1	7/5/3	0.192

BMD, bone mineral density; BMI, body mass index; BKP, balloon kyphoplasty; IVC, intravertebral cleft; NVF, new vertebral fracture; TL-Junction, the treated vertebrae located at the level of T12–L2; VF, vertebral fracture; VP, vertebroplasty.

**Table 2 diagnostics-13-03459-t002:** Logistic regression analysis of NVFs-associated variables in the training set.

	Univariate Analysis		Multivariate Analysis	
Variable	OR (95% CI)	*p*-Value	OR (95% CI)	*p*-Value
Age	1.505 (1.072, 2.113)	0.018 *	1.235 (0.795, 1.917)	0.347
Sex	0.845 (0.614, 1.164)	0.303	-	-
Smoking	0.899 (0.648, 1.247)	0.523	-	-
BMI	1.102 (0.799, 1.521)	0.553	-	-
BMD	0.874 (0.629, 1.212)	0.419	-	-
Radiomics signature	6.049 (3.415, 10.714)	<0.001 *	5.495 (3.035, 9.951)	<0.001 *
IVC	1.775 (1.267, 2.488)	0.001 *	1.524 (0.990, 2.346)	0.056
Surgical procedure	0.844 (0.609, 1.170)	0.309	-	-
Number of treated vertebra	1.160 (0.843, 1.596)	0.364	-	-
Location of treated vertebra	0.744 (0.539, 1.025)	0.071	-	-
Number of previous VF	2.041 (1.437, 2.900)	<0.001 *	1.907 (1.200, 3.031)	0.006 *

BMD, bone mineral density; BMI, body mass index; CI, confidence interval; IVC, intravertebral cleft; VF, vertebral fracture. * *p* < 0.05.

## Data Availability

The datasets utilized and/or analyzed during the current study are available from the corresponding author upon reasonable request.

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
