# Peer review of "Preoperative Prediction of New Vertebral Fractures after Vertebral Augmentation with a Radiomics Nomogram"

_diagnostics, 2023, doi:10.3390/diagnostics13223459_

Round 1

Reviewer 1 Report

Comments and Suggestions for Authors

Dear Authors

The study design is well achieved and meets the requirements regarding structure. The study's objective is clear, and the methodology is clearly described. The implications could be more striking regarding the gains from using the prediction proposal presented. The English is quite perfect and revised. The tables and figures are according to text, and they are adequate. The methodology used is highly complex, and the authors manage to expose the essential information.

Overall, good work!

Author Response

Dear Reviewer:

We highly appreciate your comments and suggestions on our manuscript entitled “Preoperative prediction of new vertebral fractures after vertebral augmentation with a radiomics nomogram”. Those comments help improve our manuscript in many aspects.

We have made revisions according to the comments with revised text highlighted in yellow in our submitted manuscript. In order to reduce unnecessary information and present more key information, we have reduced the number of figures in the main text to two and replaced the radiomics study flowchart with a figure showing cumulative incidence of NVFs.

Reviewer 2 Report

Comments and Suggestions for Authors

Dear authors

An interesting study is presented using MRI to detect the risk of postinteventional fractures for distinctive vertebrae after vertebroplasty/kyphoplasty.

Methods and results seem promising but the quality of presentation is insufficient as the reader is overloaded with unnecessary information.

Reduce the tables and figures to one each - representing the most representative ones- and destilate you findings to one major key point.

Present one key message for the reader.

Author Response

Comments 1:

Methods and results seem promising but the quality of presentation is insufficient as the reader is overloaded with unnecessary information.

Reduce the tables and figures to one each - representing the most representative ones- and destilate you findings to one major key point.

Present one key message for the reader.

Response 1:

Dear Reviewer:

We highly appreciate your comments and suggestions on our manuscript entitled “Preoperative prediction of new vertebral fractures after vertebral augmentation with a radiomics nomogram”. Those comments help improve our manuscript in many aspects.

We have made revisions according to the comments with revised text highlighted in yellow in our submitted manuscript. In order to reduce unnecessary information, we moved the radiomics study flowchart and the performance tables for individual variables to the supplementary materials. Additionally, to further highlight the clinical significance of our nomogram, we added risk accumulation curves of NVFs in the revised manuscript. In addition, some redundant words from the Methods and Results were removed.